# Increased Vulnerability to Ferroptosis in FUS-ALS

**DOI:** 10.3390/biology13040215

**Published:** 2024-03-26

**Authors:** Muhammad Ismail, Dajana Großmann, Andreas Hermann

**Affiliations:** 1Translational Neurodegeneration Section “Albrecht Kossel“, Department of Neurology, University Medical Center Rostock, University of Rostock, 18147 Rostock, Germany; muhammad.ismail@med.uni-rostock.de (M.I.); dajana.grossmann@med.uni-rostock.de (D.G.); 2German Center for Neurodegenerative Diseases (DZNE) Rostock/Greifswald, 18147 Rostock, Germany; 3Center for Transdisciplinary Neurosciences Rostock (CTNR), University Medical Center Rostock, University of Rostock, 18147 Rostock, Germany

**Keywords:** amyotrophic lateral sclerosis, oxidative damage, mitochondria, cell death

## Abstract

**Simple Summary:**

Ferroptosis is a form of regulated cell death characterized by lipid peroxide accumulation and is involved in various disease conditions, including neurodegenerative disease. However, there are still only a few reports on amyotrophic lateral sclerosis (ALS). This study addressed whether FUS-ALS-causing mutations lead to an increased vulnerability to ferroptosis. Both HeLa cells and iPSC-derived spinal motor neurons expressing ALS-causing mutations exhibited heightened vulnerability to ferroptosis-inducing agents compared to control conditions. Findings suggest that FUS mutation downregulates xCT, thus disturbing glutathione metabolism, increasing oxidative stress, and enhancing lipid peroxidation. Iron chelation, inhibition of lipid peroxidation, and mitochondrial calcium uniporter mitigated cell demise, indicating potential therapeutic targets for FUS-related ALS. The study further emphasizes the role of lipid peroxidation and ferroptosis in FUS-associated ALS.

**Abstract:**

Ferroptosis, a regulated form of cell death characterized by iron-dependent lipid peroxide accumulation, plays a pivotal role in various pathological conditions, including neurodegenerative diseases. While reasonable evidence for ferroptosis exists, e.g., in Parkinson’s disease or Alzheimer’s disease, there are only a few reports on amyotrophic lateral sclerosis (ALS), a fast progressive and incurable neurodegenerative disease characterized by progressive motor neuron degeneration. Interestingly, initial studies have suggested that ferroptosis might be significantly involved in ALS. Key features of ferroptosis include oxidative stress, glutathione depletion, and alterations in mitochondrial morphology and function, mediated by proteins such as GPX4, xCT, ACSL4 FSP1, Nrf2, and TfR1. Induction of ferroptosis involves small molecule compounds like erastin and RSL3, which disrupt system Xc^−^ and GPX4 activity, respectively, resulting in lipid peroxidation and cellular demise. Mutations in fused in sarcoma (*FUS*) are associated with familial ALS. Pathophysiological hallmarks of FUS-ALS involve mitochondrial dysfunction and oxidative damage, implicating ferroptosis as a putative cell-death pathway in motor neuron demise. However, a mechanistic understanding of ferroptosis in ALS, particularly FUS-ALS, remains limited. Here, we investigated the vulnerability to ferroptosis in FUS-ALS cell models, revealing mitochondrial disturbances and increased susceptibility to ferroptosis in cells harboring ALS-causing FUS mutations. This was accompanied by an altered expression of ferroptosis-associated proteins, particularly by a reduction in xCT expression, leading to cellular imbalance in the redox system and increased lipid peroxidation. Iron chelation with deferoxamine, as well as inhibition of the mitochondrial calcium uniporter (MCU), significantly alleviated ferroptotic cell death and lipid peroxidation. These findings suggest a link between ferroptosis and FUS-ALS, offering potential new therapeutic targets.

## 1. Introduction

Ferroptosis is a regulated form of cell death induced by the accumulation of lipid peroxides, a process dependent on the cellular iron levels [1]. This process involves the excessive accumulation of reactive oxygen species (ROS), particularly lipid peroxides, leading to oxidative death [1,2,3]. Key features of ferroptosis include its dependency on intracellular iron level, the depletion of the antioxidant glutathione (GSH) level, oxidative stress, and alterations in mitochondrial membrane potential and function [1,3].

A number of proteins, including glutathione peroxidase 4 (GPX4), cystine-glutamate transporter (system Xc^−^), ferroptosis suppressor protein-1 (FSP1), nuclear factor erythroid 2-related factor 2 (Nrf2), and Transferrin receptor protein 1 (TfR1), regulate ferroptosis by influencing both iron processing and lipid peroxidation [1,4,5,6,7]. Ferroptotic cell death can be induced by synthetically designed compounds, such as erastin or RSL3 [8,9]. However, erastin directly inhibits the glutamate/cystine antiporter system Xc^−^ (xCT), which is the uppermost regulator in the molecular pathway of ferroptosis [8,10].

Inhibition of the xCT results in the depletion of cysteine, which causes a further reduction in GSH synthesis as cysteine is a constructive element in GSH synthesis [1,8,10]. Consequently, the reduction in GSH levels leads to a decreased antioxidant capability, resulting in increased ROS levels and thereby inducing non-apoptotic cell death [11,12]. GSH depletion hinders the activity of antioxidant enzymes, such as GPX4, and ultimately leads to ferroptotic cell death. In addition, targeting GPX4 activity using RSL3 induces the accumulation of lipid peroxides and ferroptosis [1,3,6].

Consequently, various compounds have been formulated, including liproxstatin-1 (Lip-1) and ferrostatin-1 (fer-1), which inhibit ferroptosis by scavenging lipid peroxyl radicals formed during lipid peroxidation [13,14]. Additionally, as ferroptosis relies on the presence of iron, studies have shown the iron chelator deferoxamine (DFO) to inhibit ferroptosis [1,13,15]. Notably, abnormalities of the iron metabolism have been associated with neurodegenerative diseases for many years, and iron chelators such as DFO are the subjects of clinical trials in various neurodegenerative diseases.

Hallmarks of ferroptosis involve changes in mitochondrial structure and function. These changes encompass mitochondrial fragmentation, disruption of the mitochondrial membrane potential (ΔΨ_m_), elevated levels of mitochondrial calcium ([Ca^2+^]_m_), and increased production of mitochondrial ROS [13,16,17,18]. Calcium ions are essential for many cellular functions, such as regulating signaling pathways and initiating cell-death processes [19]. Alteration in cytosolic Ca^2+^ levels can impact cell function and trigger cellular responses, such as ferroptosis induction [18,20]. The mitochondrial calcium uniporter (MCU) serves as the main entry point of Ca^2+^ into the mitochondrial matrix. This important function makes MCU crucial for regulating the mitochondrial calcium level in cells [19,21]. Inhibition of the MCU with, e.g., Ru265, a modified ruthenium compound, is known to exhibit potential neuroprotective effects against hypoxic/ischemic (HI) brain injury. A recent study reported that different inhibitors of the MCU, including Ru265, were also able to alleviate erastin-mediated ferroptotic cell death in diverse cell types. However, the potential anti-ferroptotic effects of MCU inhibition in neurodegeneration, and particularly in FUS-ALS, have not been evaluated so far.

Amyotrophic lateral sclerosis (ALS) is a progressive and ultimately fatal neurodegenerative disease marked by the deterioration of upper and lower motor neurons in regions like the motor cortex, brainstem, and spinal cord, typically leading to mortality within 2–5 years of symptom onset [22,23]. While the majority of ALS cases are sporadic, about 10–15% are caused by mutations in several genes including Superoxide dismutase 1 (*SOD1*), Tar DNA binding protein (*TARDBP*), Chromosome 9 open frame 72 (*C9ORF72*), and Fused in Sarcoma (*FUS*) [23,24]. *FUS* mutations account for the third most common genetic variant, contributing to about 5% of familial ALS cases and up to 1% of sporadic ALS (sALS) cases, but even more in juvenile onset cases [25]. While the pathophysiology of ALS is not yet fully understood, various cellular and molecular processes were reported to be disturbed, including glutamatergic excitotoxicity, mitochondrial dysfunction, and oxidative stress [26,27,28], all of which are hallmarks of ferroptosis. While ferroptosis has been investigated in many disease conditions, including Parkinson’s disease, surprisingly few studies exist on ferroptosis in ALS. Genetic depletion of GPX4, a central repressor of ferroptosis, induced motor phenotypes suggestive of motor neuron diseases accompanied by a more or less selective motor neuron degeneration [29]. GPX4 was reported to be reduced in spinal cord lysates of sporadic and familial ALS patients [23]. GPX4 overexpression in hSOD1G93A mice resulted in delayed disease onset and prolonged survival [23]. In hSOD1-G93A, overexpressing cells and mice lipid peroxidation was reported to be induced by TFR1-imported excess free iron, accompanied by decreased GSH and mitochondrial membrane dysfunction [30]. No data exist so far for FUS-ALS, despite the fact that FUS-ALS pathophysiology is particularly characterized by oxidative and mitochondrial damage [31].

We thus aimed to investigate the vulnerability to ferroptosis in FUS-ALS using Hela cells, which had been previously edited to carry BACs with either WT-FUS-eGFP or FUS-P525L-eGFP [32]. We systematically investigated cell vulnerability to ferroptosis inducers and inhibitors, as well as the cellular ROS level, and lipid peroxidation in wild-type and FUS-ALS mutant cells, including iPSC-derived motor neurons. Overall, our study not only highlights the heightened vulnerability of FUS-ALS cells to induced ferroptosis but also provides insights into the potential molecular mechanisms underlying this susceptibility through comprehensive analysis of lipid peroxide levels, expression profiles of ferroptosis-associated proteins and glutathione levels.

## 2. Material and Methods

### 2.1. HeLa Cell Culture

HeLa cell lines with either WT-FUS-eGFP or FUS-P525L-eGFP were grown in Dulbecco’s modified Eagle’s medium (DMEM), containing 10% FBS, 1% penicillin/streptomycin/glutamine (Thermo Fisher Scientific, Waltham, MA, USA). All the experimental cells were tested on a monthly basis for Mycoplasma contamination using the Plasmo Test_Detection Kit (InvivoGen).

### 2.2. Differentiation of iPSC-Derived Motor Neurons (MN)

The generation of MNs from human neural precursor cells (NPCs), which had been previously derived from human induced pluripotent stem cells (iPSCs) was accomplished following the protocol from published data [33]. Human NPCs, an isogenic CRISPR-Cas cell line carrying the P525L-mutated FUS, and the WT-FUS generated and characterized previously [33] were cultured on 1:100 matrigel coated dishes in N2B27 medium containing 48.5% DMEM/F12 (Thermo Fisher Scientific, Waltham, MA, USA), 48.5% Neurobasal medium (Invitrogen, Carlsbad, CA, USA), 1% penicillin/streptomycin/glutamine (Thermo Fisher Scientific, Waltham, MA, USA), 1% B27 supplement without vitamin A (Invitrogen, Carlsbad, CA, USA), and 0.5% N2 supplement (Invitrogen, Carlsbad, CA, USA) with the addition of 3 µM CHIR99021 (Cayman chemical company, Ann Arbor, MI, USA), 150 µM ascorbic acid (Sigma Aldrich, Chemie GmbH, Munich, Germany), and 0.25 µM purmorphamine (Cayman chemical company, Ann Arbor, MI, USA). MN differentiation was as follows: NPCs were re-seeded with a density of 100,000 cells per well of a 6-well plate, PLO/laminin-coated. Differentiation was started using an N2B27 medium also containing 1 ng/mL rhBDNF (Promega, Madison, WI, USA), 1 µM retinoic acid (Sigma Aldrich, St. Louis, MO, USA), 200 μM ascorbic acid (Sigma Aldrich, Chemie GmbH, Munich, Germany), 1 µM purmorphamine, and 1 ng/ mL rhGDNF (Sigma Aldrich, St. Louis, MO, USA). Maturation medium (9 days later) consisted of N2B27 medium supplemented with 100 µM DBcAMP (Sigma Aldrich, St. Louis, MO, USA), 2 ng/mL rhBDNF, 1 ng/mL TGFβ3 (AF-100-36E; Thermo Fisher Scientific; Waltham, MA, USA), 200 μM ascorbic acid, and 2 ng/mL rhGDNF (Sigma Aldrich, St. Louis, MO, USA). Medium was changed every second day. MN analysis and viability experiments were performed after 3 weeks in maturation conditions unless otherwise indicated.

### 2.3. Live-Cell Imaging

For analysis of the mitochondrial membrane potential (MMP) and lipid peroxidation, FUS-eGFP HeLa cells were seeded into 8-well µ-slides at a density of 10,000 cells per well (Ibidi: 80806-90, Gräfelfing, Germany). Cells were grown in DMEM + 10% FBS + 1% Pen/Strep for 24 h. On the day of the MMP measurement, cells were stained with 50 nM Tetramethylrhodamine Ethyl Ester (TMRE; Invitrogen: T669, Schwerte, Germany) and 1 nM of MitoTracker Deep Red (Invitrogen: M22426, Schwerte, Germany) for 30 min at 37 °C. Live-cell imaging was performed using Zeiss inverted AxioObserver with LSM 900 module, 10× and 63× high-resolution objective, with oil at 37 °C and 5% CO_2_ incubation. 

Mitochondrial ROS was analyzed using fluorescence probe MitoSOX Red (MRS) (Thermo Fisher, M36008, Waltham, MA, USA). MitoSOX Red was used to selectively stain mitochondrial superoxide production, and for costaining, we used MitoTracker deep red (Thermo Fisher Scientific, M22426, Waltham, MA, USA). Cells were cultured in DMEM + 10% FBS + 1% Pen/Strep for 24 h. On the day of the mitochondrial lipid peroxides level measurement, cells were stained with 100 nM MRS and 1 nM of MitoTracker Deep Red for 30 min at 37 °C. Live-cell imaging was performed using Zeiss inverted AxioObserver with LSM 900 module, 63× high-resolution objective, with oil at 37 °C and 5% CO_2_ incubation. Data acquired from live-cell imaging were evaluated using macros in Fiji, powered by ImageJ (Version v1.53). Mask are depicted in the respective figures.

The level of cellular lipid peroxidation was assessed using the fluorescent dye boron-dipyrromethene BODIPY 665/676 (Thermo Fisher, B3932, Waltham, MA, USA). On the following day, the growth medium containing different compounds was replaced with equal volumes of normal growth medium containing 2 µM BODIPY 665/676, and cells were incubated for 1 h at 37 °C. Live-cell imaging was conducted on a Zeiss inverted AxioObserver with LSM 900 module, using 10× resolution objective, and full environment control of 37 °C and 5% CO_2_. 

For analysis of the lipid peroxidation, cells were treated with specific concentrations of different compounds, such as (1S, 3R)-RSL3 (Merck, SML2234, Darmstadt, Germany), erastin (Merck, 329600-5M), and Ru265 (Merck/Sigma-Aldrich, SML2991, Darmstadt, Germany) for 24 h. LSM analysis revealed a significant increased production of ROS within the mitochondria of Hela-FUS-P525L cells compared to Hela-FUS-WT cells using fluorescence probe 2 µM BODIPY 665/676, which selectively stains lipid peroxides. Live-cell imaging was conducted on a Zeiss inverted AxioObserver with LSM 900 module, using 10× resolution objective, and full environment control of 37 °C and 5% CO_2_. Image analysis was performed using Fiji, powered by ImageJ (Version v1.53). Data acquired from live-cell imaging were evaluated using macros in Fiji.

### 2.4. PrestoBlue Cell Viability Assay

To analyze cell viability, cells were cultured into 96-well plates (735-0465, VWR) at a density of (2000 cells/well) in 100 μL of medium per well in triplicate per each group treatment and incubated overnight at 37 °C with 5% CO_2_. The next day, cells were treated with increasing concentrations of different compounds, including (1S, 3R)-RSL3, erastin with and without liproxstatin-1 (Lip-1, Merck, SML1414, Darmstadt, Germany), and Deferoxamine (DFO, Merck, D9533, Darmstadt, Germany), for 24 h. The cell viability was analyzed using PrestoBlue Viability Assay (Thermo Fisher Scientific) according to the protocol provided by the manufacturer. Resazurin is the active ingredient of PrestoBlue reagent, which converted to resorufin upon the reducing environment of living cells. Conversion of resazurin to resorufin induced a color change to light pink. PrestoBlue was diluted to 1:10 ratio with culture medium and incubated with the cells at 37 °C with 5% CO_2_ for 15 min. Absorbance (560 nm excitation and 590 nm emission) was read on Microplate Reader (Spark^R^, Tecan, Switzerland) after 15 min incubation at 37 °C with 5% CO_2_. Cell viability measurements of each line were normalized to the average values of untreated wells of each cell line.

### 2.5. Measurement of GSH Levels

A luciferase-based assay was used to analyze total glutathione (GSH) and oxidized glutathione (GSSG) levels in the experimental cell lines. The assay was performed following the protocol of Promega^®^ GSH/GSSG Assay ( Madison, WI, USA). Cells were seeded into 96-well plate (735-0465, VWR) at a density of (2000 cells/well) in 100 μL of medium per well. The no-cells control was used to normalize the background signal from the assay chemistry, which was then subtracted from vehicle and test signals to give net values. Absorbance (555 nm excitation and 565 nm emission) was measured on Microplate Reader (Spark^R^, Tecan, Switzerland) with in an incubation time of 15 min. 

### 2.6. Immunoblotting

Cells were harvested using trypsinization and then subjected to centrifugation at 4 °C for 15 min at maximum speed. The resulting supernatant was carefully transferred to a fresh tube and stored at −20 °C until needed. Following this, the cells were rinsed with 1× PBS solution and subsequently lysed using RIPA buffer. The composition of the RIPA buffer included 125 mM NaCl, 25 mM Tris-Cl (pH 7.4), 1% Triton X-100, 0.5% sodium deoxycholate, 0.1% SDS, and 1× complete EDTA-free protease inhibitors. The protein amount was determined using the BCA assay from Thermo Fisher Scientific. Subsequently, immunoblot analysis was carried out using a 12% SDS gel, which was then transferred to a ready-to-use Trans-Blot^®^ Turbo™ PVDF membrane from Bio-Rad. The transfer of proteins was accomplished using the standard program (30 min, 25 volts, 1 ampere) in the semi-dry Trans-Blot^®^ Turbo™ Transfer System from Bio-Rad (Hercules, CA, USA). After that, the blotting cassette was disassembled, and the membranes were placed in a chamber and washed three times with TBS-T before blocking was started. The membranes were then incubated under constant shaking at room temperature for 1 h in blocking buffer (5% non-fat milk) dissolved in Tris-Buffered Saline supplemented with 0.1% Tween-20 (TBS-T). Next, the membranes were incubated with a primary antibody diluted according to manufacturer’s protocol in the corresponding blocking solution for an overnight incubation at 4 °C on a shaker. The next day, three washes were carried out with TBS-T for 5 min at room temperature. Following the incubation with the secondary antibody, which was diluted according to the manufacturer’s protocol, the membrane was placed on a shaker at room temperature for 1 h. Subsequently, the membranes underwent an additional 3x washes with TBS-T for 5 min each. For visualization, the membrane was wrapped in transparent plastic wrap and treated with Clarity™ Western ECL Blotting Substrate from Bio-Rad for 1 min. Excess reagent was then removed, and the blot was scanned utilizing an imaging system from LI-COR Biosciences (NE, USA). To redevelop the membrane with a different antibody, the antibodies were stripped from the PVDF membrane using 0.4 M NaOH for 7 min. After this process, the protein bands were quantified with Image Lab software from Bio-Rad. Data obtained from live-cell imaging were analyzed using Fiji, powered by ImageJ. Statistical analysis was conducted using GraphPad Prism 8 software. 

### 2.7. Statistics

All experiments, including dose-dependent cell viability, enzymatic assays, and Western blots, were performed in triplicate and independently repeated at least three times, respectively (*n* represents the number of independent biological replicates), unless otherwise stated. Data were collected and analyzed using GraphPad Prism 8.0 software. For statistical tests, we used a two-way ANOVA and performed a Tukey post hoc test for comparison of multiple experimental groups. The presented data represent mean values with standard deviation (±s.d.). Statistical significance is indicated with asterisks: * *p* < 0.05, ** *p* < 0.01, *** *p* < 0.001, **** *p* < 0.0001, unless otherwise stated in the figure legend. 

## 3. Results

### 3.1. Mitochondrial Depolarization in FUS-Mutated Cells

Mitochondrial dysfunction is a characteristic of ferroptosis. FUS-ALS pathophysiology was associated with significant mitochondrial dysfunction. Specifically, axonal mitochondrial transport was shown to be impaired, as well as mitochondrial depolarization, which was obvious when observed together with the impacted metabolic state of FUS-mutated neurons [33,34]. We first investigated whether this mitochondrial phenotype is also seen in HeLa cells engineered to carry either wild-type FUS-eGFP or P525L FUS-eGFP. Therefore, we investigated whether the inner membrane potential, known as a crucial indicator of mitochondrial function, is disrupted in FUS-mutated HeLa cell lines.

For this, we used Tetramethylrhodamine ethyl ester (TMRE), a cell-permeable fluorescent dye that accumulates in active mitochondria in comparison to Mitotracker deep red. Hela-FUS-P525L cells showed an almost halved membrane potential compared to Hela-FUS-WT cells (Figure 1), thereby confirming the mitochondrial dysfunction being specifically associated with FUS-ALS across different cell types. 

### 3.2. FUS Mutation Results in Increased Production of Reactive Oxygen Species

Reactive oxygen species (ROS), produced as natural byproducts of cellular metabolism, are highly reactive molecules containing oxygen [35]. Overproduction of ROS can lead to oxidative stress, which can harm cellular components, including DNA, proteins, and lipids [36]. Therefore, we aimed to measure and understand how FUS mutation influences the generation of ROS, particularly within the mitochondria of the cells. To analyze mitochondrial lipid peroxides, we used the fluorescence probe MitoSOX, which selectively stained mitochondrial superoxide production, and costained with MitoTracker deep red. LSM analysis revealed a significant increased production of ROS within the mitochondria of Hela-FUS-P525L cells compared to Hela-FUS-WT cells (Figure 2). These results thereby confirm that the mitochondrial ROS generation in HeLa is specifically associated with FUS-ALS.

### 3.3. Increased Sensitivity to Ferroptosis in FUS-Mutated Cells

Ferroptosis leads to a reduction in cell viability via inducing oxidative cell death [1]. Having shown mitochondrial depolarization and ROS production in FUS-ALS mutants, we hypothesized that ferroptosis might be significantly involved in FUS-ALS pathophysiology. Glutathione peroxidase 4 (GPX4) mediates lipid peroxidation and is thus a key enzyme of ferroptosis [37], whereas system Xc^−^ cystine/glutamate antiporter (xCT) plays a key role in controlling the intracellular redox state. Inhibition of the latter leads to increased ROS production upstream of GPX4 [38].

To investigate the vulnerability of FUS-mutated cells to ferroptosis, we conducted a comprehensive investigation into the dose-dependent response of cell death induced by RSL3 and erastin, two well-known ferroptosis inducers either interfering with GPX4 or xCT, respectively. Following treatment with varying concentrations of RSL3 and erastin for a duration of 24 h, both HeLa-FUS-P525L and HeLa-FUS-WT cells exhibited a notable change in morphology suggestive of cell death (Figure 3A,B) and cell viability loss in a dose-dependent manner (Figure 3C,D). Cell morphology changes and cell proliferation were significantly reduced in the treated cells, particularly in HeLa-FUS-P525L (Figure 3C,D). Of note, HeLa-FUS-P525L cells were significantly more vulnerable to induced ferroptosis compared to HeLa-FUS-WT cells (Figure 3C,D). All of these were abolished in case of co-treatment with the ferroptosis inhibitor Lip-1 (Figure 3A–D). These findings suggest that FUS mutated HeLa cells exhibit heightened vulnerability to key ferroptosis inducers. 

### 3.4. FUS Mutated Cells Exhibit Misregulation of Key Factors of Ferroptosis

GPX4 knockout studies revealed motor neuron disease phenotypes as a consequence of (selective) motor neuron loss [29]. Furthermore, a reduced level of GPX4 has been reported in spinal cord lysates of sporadic and familial ALS patients [23]. To further elucidate possible explanations for the increased vulnerability to ferroptosis in FUS-ALS, we next conducted an analysis of the expression of the main ferroptosis proteins GPX4, xCT, ACSL4, and FSP1 (Figure 4A,B).

Significantly increased expression of FUS proteins was due to FUS mutation (Figure 4B). Interestingly, the immunoblot analysis indicated only a non-significant decrease in the protein expression levels of GPX4, but a significant reduction in xCT and FSP1 expression, while there was an increase in ACSL4 protein expression in HeLa-FUS-P525L cells compared to HeLa-FUS-WT cells (Figure 4A,B). 

### 3.5. FUS Mutations Lead to Disturbance of Cellular Redox Defense System Downstream of xCT

Having shown that xCT is significantly reduced in FUS-ALS cells, we analyzed this pathway further. xCT plays a crucial role in facilitating the cellular uptake of cystine, which is necessary for the synthesis of GSH, an important antioxidant in cells [1]. Ferroptosis, as a regulated form of cell death, relies on both iron and ROS level. Free iron, a strong oxidant, can generate highly reactive hydroxyl radicals, especially through the Fenton reaction. DFO, which is categorized as an iron chelator, has gained attention for its ability to reduce the availability of free iron within the intracellular environment. By curtailing the Fenton reaction, a pivotal mechanism driving lipid peroxidation and consequent cell demise by ferroptosis, DFO has shown promise for the reduction of ferroptotic cell death in various studies [39]. We thus asked whether DFO is able to restore ferroptosis in FUS-ALS cells as well. We conducted co-treatment experiments using both HeLa-FUS-P525L and HeLa-FUS-WT cells (Figure 4). Our results revealed that the induction of ferroptotic cell death in both cell lines by variable concentrations of erastin was prevented when treated with 100 µM DFO (Figure 4C,D). Not only do these outcomes underline the significant contribution of ferroptosis to cell death observed in FUS-ALS, but they could also offer a new treatment option. 

Downregulation of xCT results in decreased cellular uptake of cystine, leading to reduced synthesis of GSH. Glutathione is a key antioxidant that helps neutralize ROS and protects cells from oxidative damage [40]. However, when cellular levels of reduced glutathione are depleted or compromised, the balance between oxidants and antioxidants shifts towards oxidative stress, leading to the accumulation of lipid peroxides [41]. GSSG accumulation due to imbalanced redox conditions can contribute to increased susceptibility to lipid peroxidation [42]. We thus evaluated the level of GSH by comparatively analyzing the cellular GSH level in both HeLa-FUS-P525L and HeLa-FUS-WT cells (Figure 5A). The obtained data revealed a significant reduction in the level of oxidized glutathione (GSSG) in HeLa-FUS-P525L cells. Although GSSG reduction itself does not directly induce lipid peroxidation, its accumulation due to imbalanced redox conditions can contribute to increased susceptibility to lipid peroxidation.

The accumulation of lipid peroxidation products within the cellular membrane can lead to membrane disruption and ultimately to cell death mediated by ferroptosis. Lipid peroxidation, a marker of oxidative stress, indicates an imbalance between the production of ROS and the cell’s antioxidant defense mechanisms [43]. We thus investigated whether there is increased level of lipid peroxidation in the case of FUS mutants. Here we utilized the lipid peroxidation sensor, BODIPY 665/676, a commercially available fluorescent dye, to assess lipid peroxide accumulation in living cells [44] (Figure 5B). BODIPY 665/676 demonstrates altered fluorescence upon interaction with peroxyl radicals. Laser scanning microscopy (LSM) revealed higher levels of lipid peroxides that accumulate primarily in the plasma membrane of Hela-FUS-P525L cells compared to Hela-FUS-WT (Figure 5B,C). Next, we analyzed the intensity of lipid peroxides in both cell lines following the treatment with xCT inhibitor erastin of 20 μM [1] (Figure 5D,E). Surprisingly, the inhibition of xCT with 20 μM of erastin did not result in a significant further increase in lipid peroxidation compared to untreated cells (Figure 5D,E).

### 3.6. Inhibition of Mitochondrial Calcium Uniporter Alleviates Lipid Peroxidation

Inhibition of xCT was reported to lead to mitochondrial fragmentation, mitochondrial calcium overload, increased mitochondrial ROS production, and disruption of the mitochondrial membrane potential [21]. The observation of mitochondrial dysfunction, as a hallmark of ferroptosis (Figure 1A,B), makes preservation of mitochondrial function a potential therapeutic strategy against ferroptotic cell death. Calcium overload has been identified as playing a significant role in the cell-death mechanism, including ferroptosis [18,21].

Mitochondrial calcium levels are controlled via the MCU, which represents the main entry point of Ca^2+^ into the mitochondrial matrix [19,21]. A recent report suggested that inhibition of the MCU might be protective against lipid peroxidation and ferroptosis in wild-type cells [21]. Having shown both mitochondrial dysfunction (Figure 1) and increased vulnerability to ferroptosis (Figure 2 and Figure 3), we hypothesized that inhibition of the MCU might also be protective in FUS-ALS cell lines. The ruthenium compound Ru265 is a commonly studied pharmacological inhibitor of the MCU and inhibitor of lipid peroxide generation [21,45]. To this end, we treated cells with erastin with or without Ru265. A significantly reduced intensity of lipid peroxides was observed in the cells co-treated with 20 µM erastin and 50 µM Ru265 (Figure 5D), while 100 µM Ru265 increased cell death in FUS mutants (Figure 5E). Co-treatment of Ru265 with erastin appeared to decrease cellular lipid peroxides in both wild-type and FUS-mutated cell lines. These results demonstrated that FUS mutations lead to increased vulnerability to ferroptosis, most likely by impairing cellular redox signaling downstream of xCT. Both iron chelation and interfering with the MCU might be interesting targets for the alleviation of ferroptosis in FUS-ALS.

### 3.7. Increased Vulnerability of FUS-Mutated Human Motor Neurons to Ferroptosis

Having shown increased vulnerability to ferroptosis in a non-neuronal FUS-ALS model system, we finally asked whether similar effects are seen in human motor neurons. For this purpose, we used spinal motor neurons with FUS mutations generated from FUS-mutated human-induced pluripotent stem cell-derived neural progenitor cells (NPCs) and isogenic CRISPR/Cas9-generated cell lines carrying either WT-FUS-eGFP or the P525L-eGFP mutant (Figure 6) [33]. We induced ferroptosis using a variable concentration of erastin and RSL3 to compare the vulnerability of FUS-mutated motor neurons (MN-FUS-P525L) to wild-type motor neurons (MN-FUS-WT). Following the treatment of both MN-FUS-P525L and MN-FUS-WT using variable concentrations of erastin and RSL3 for 48 h, a notable alteration in morphology was observed, suggesting cell death (Figure 5A). Cell viability assays revealed a notable difference in susceptibility to both erastin and RSL3-induced ferroptosis between MN-FUS-P525L and MN-FUS-WT (Figure 5B,C), with a particular vulnerability to erastin and RSL3-induced ferroptosis in FUS-P525L motor neurons compared to FUS-WT motor neurons. These findings suggest an increased vulnerability to ferroptosis for FUS-ALS disease-causing mutations in different cell types, including human motor neurons.

## 4. Discussion

The results of our study reveal compelling evidence indicating an association between FUS mutations and heightened susceptibility to ferroptosis, a regulated form of necrotic cell death characterized by the accumulation of lipid peroxides [1]. Our results further suggest that downregulation of xCT due to FUS mutation led to disturbed GSH metabolism and increased lipid peroxidation, which could be alleviated by either iron chelation or MCU inhibition.

Ferroptosis, characterized by the reduction in cell viability due to oxidative cell death, is governed by key enzymes such as glutathione peroxidase 4 (GPX4) and the cystine/glutamate antiporter system Xc^−^ [10]. Understanding the cellular mechanisms underlying ferroptosis susceptibility in the context of FUS mutations is crucial for advancing therapeutic interventions in FUS-related neurodegenerative diseases. To investigate the vulnerability of FUS mutated cells to ferroptosis, we utilized a model system of HeLa cells expressing either a FUS-P525L mutant or a FUS-WT Our comprehensive investigation into the dose-dependent response of cell death induced by the ferroptosis inducers RSL3 and erastin revealed a notable induction of cell death in both FUS-mutated and wild-type cells in a dose-dependent manner (Figure 3). Previous research by Do Van et al. (2016) and Ayala and Muñoz (2019) has implicated ferroptosis in various neurodegenerative disorders, including ALS [46,47]. They have shown that oxidative stress and lipid peroxidation, as hallmarks of ferroptosis contribute to neuronal death and disease pathology. Interestingly, FUS-mutated HeLa cells (FUS-P525L HeLa) and human-induced pluripotent stem cell-derived motor neurons (MN-FUS-P525L) exhibited significantly compromised cellular viability compared to FUS-WT HeLa cells (Figure 3C,D) and MN-FUS-WT (Figure 6B,C) to both erastin and RSL3, which supports the notion that ferroptosis plays a relevant role in FUS-related neurodegeneration (Figure 3C,D). Liproxstatin-1 (Lip-1) is widely recognized as an inhibitor of ferroptosis [2]. Co-treatment experiments revealed that cell death induced by varying concentrations of erastin and RSL3 was completely inhibited in the presence of Lip-1, further supporting ferroptosis as the underlying cell-death mechanism (Figure 3C,D). This differential susceptibility to ferroptosis suggests that FUS mutations may predispose cells to heightened oxidative stress and lipid peroxidation, leading to increased susceptibility to ferroptosis-mediated cell death.

As mentioned above, the regulation of key proteins involved in ferroptosis, such as xCT and GPX4, plays a crucial role in determining the susceptibility of cells to this form of cell death [1,37]. Interestingly, we noted a reduction in xCT as the most misregulated finding in the case of FUS-ALS (Figure 4A,B). FSP1 was also significantly reduced in FUS-ALS, whereas GPX4 was not changed significantly (Figure 4A,B). Of note is that we also observed a significant increased FUS level in cases of FUS mutation (Figure 4A,B). It is indeed known that FUS protein levels are increased in cases of FUS mutation. It was shown previously that FUS autoregulates its own protein level by binding to exon 7 of its own transcript, leading to alternative splicing due to exon 7 skipping. Consequently, transcripts that are lacking exon 7 are degraded via nonsense-mediated decay. Mutations in FUS alter this autoregulatory mechanism, leading to accumulation of FUS protein [48,49]. Future studies should address whether wild-type FUS overexpression can phenocopy above the described phenotypes.

This notable decrease in xCT protein expression led us to hypothesize that the compromised xCT can result in decreased cellular uptake of cystine, leading to reduced synthesis of GSH, a key antioxidant involved in neutralizing ROS and protecting cells from increased oxidation [40] (Figure 5A). We found a significant reduction in the level of GSSG in HeLa-FUS-P525L cells compared to HeLa-FUS-WT cells (Figure 5A). Although GSSG depletion does not have a direct effect on inducing lipid peroxidation, the observed reduction in GSSG levels suggests a depletion or compromise in cellular levels of reduced glutathione, leading to imbalanced redox conditions. When cellular levels of reduced glutathione are depleted or compromised, the balance between oxidants and antioxidants shifts towards oxidative stress, leading to the accumulation of lipid peroxides [41]. Here we assume that xCT downregulation can possibly aggregate imbalanced redox conditions and subsequently induce ferroptotic cell death. Similar to our findings, Sato et al. (2018) and Lewerenz et al. (2013) have demonstrated the critical role of xCT in maintaining cellular redox homeostasis by regulating the import of cystine for glutathione synthesis [50,51]. This is in line with the finding that the FUS mutation itself was already sufficient to show increased lipid peroxidation, which was not further increased by xCT inhibition (Figure 5D,E). Polyunsaturated fatty acids (PUFAs) have been implicated in various neurodegenerative diseases, including ALS [52]. ACSL4 is known to promote the incorporation of polyunsaturated fatty acids (PUFAs) into phospholipids, rendering cells more susceptible to lipid peroxidation and ferroptosis [7]. Thus, the elevated ACSL4 expression in HeLa-FUS-P525L cells may have possibly contributed to the increased levels of lipid peroxides observed in these cells.

The pathophysiology of FUS-ALS significantly involves oxidative and mitochondrial damage. Interestingly, inhibition of xCT was reported to lead to diverse mitochondrial phenotypes, including mitochondrial fragmentation, excessive mitochondrial calcium accumulation, increased production of mitochondrial ROS, and disruption of the mitochondrial membrane potential [21], which we also found in FUS-mutated cells (Figure 1 and Figure 2). A recent report suggested that inhibition of the MCU might be protective against lipid peroxidation and ferroptosis in wild-type cells [21]. Excitingly, MCU inhibition was able to alleviate lipid peroxidation in both wild-type and FUS-ALS cells (Figure 5D,E). The observed protective effects of MCU inhibition against lipid peroxidation provide further support for targeting mitochondrial dysfunction in ALS therapy.

Our study has some limitations. These include, e.g., the use of a non-neuronal model system. We believe, however, that the data support the notion of an increased vulnerability to ferroptosis in cases where ALS associated FUS mutations are expressed. Nevertheless, human iPSC-derived MNs carrying the ALS-causing FUS mutation also showed increased vulnerability to ferroptosis (Figure 6), further hinting towards a relevant role in neurodegeneration. Additionally, we did not monitor cellular uptake of erastin and RSL3 to evaluate whether the cellular response is adequate for the uptake. We cannot rule out that the differential vulnerabilities observed might be due to differential uptake of the inhibitors, which deserves future investigation. However, the already increased level of ROS (Figure 2) and lipid peroxidation (Figure 5) in untreated conditions also point towards ferroptosis induction in case of FUSALS causing mutations.

## 5. Conclusions

In conclusion, our study sheds light on the role of lipid peroxidation and ferroptosis in the pathogenesis of FUS-associated ALS. The observed accumulation of lipid peroxides and dysregulation of ferroptosis-associated proteins, coupled with heightened vulnerability to ferroptotic cell death in cells expressing the FUS-P525L mutant, highlight the potential involvement of ferroptosis in FUS-related neurodegenerative diseases. In addition, the protective effects of Ru265, Lip-1, and DFO against ferroptotic cell death indicate that modulating lipid peroxidation and iron availability could mitigate cell death in FUS-associated ALS. However, further research is required to elucidate the intricate molecular mechanisms underlying ferroptosis and validate the efficacy of targeted interventions in FUS-ALS. These efforts might pave the way for the development of novel therapeutic approaches aimed at modulating ferroptosis pathways and improving outcomes for patients with FUS-related neurodegenerative diseases.

## Figures and Tables

**Figure 1 biology-13-00215-f001:**
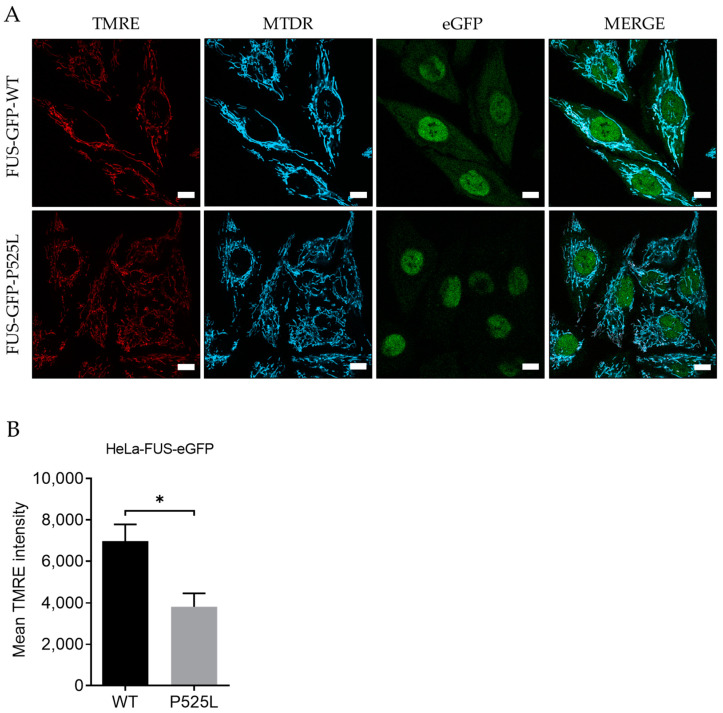
Mitochondrial depolarization in FUS-mutated cells. (**A**) HeLa cells were stained with TMRE and MitoTracker deep red for live-cell imaging. Scale bar = 10 µm (**B**) The mean TMRE signal intensity was quantified using MitoTracker deep red as counterstain for detection of mitochondria. Data are from 6 independent biological replicates (*n* = 6). See also Appendix A. Data indicated as mean ± s.d. * *p* < 0.05. (two-way ANOVA).

**Figure 2 biology-13-00215-f002:**
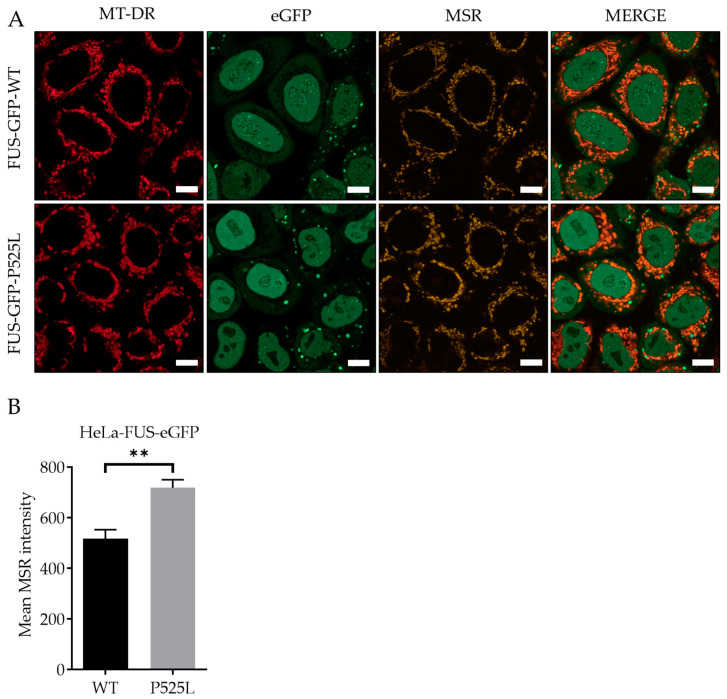
Increased production of reactive oxygen species within the mitochondria of FUS HeLa-mutated cells. Scale bar = 10 µm (**A**) HeLa cells were stained with MitoSOX Red (MSR) and MitoTracker deep red for live-cell imaging. (**B**) The mean MSR signal intensity was quantified using MitoTracker deep red as counterstain for detection of mitochondrial ROS. Data shown are from 3 independent biological replicates (*n* = 3). Data indicated as mean ± s.d. ** *p* < 0.01. (two-way ANOVA), from three independent experiments.

**Figure 3 biology-13-00215-f003:**
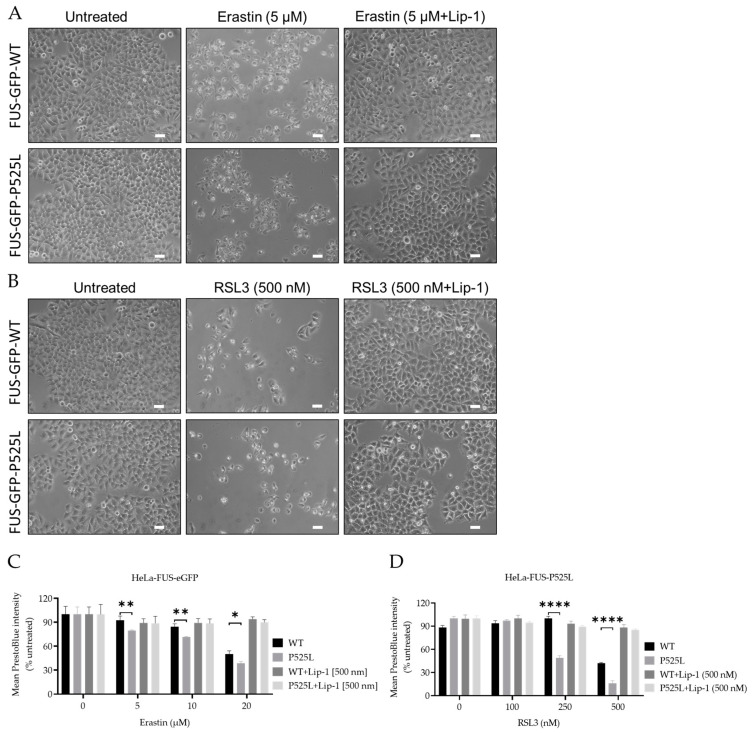
Inhibition of GPX4 and xCT enhances the vulnerability to ferroptosis, particularly in FUS-mutated cells. (**A**,**B**) Microscope images of both HeLa-FUS-WT-eGFP and HeLa-FUS-P525L-eGFP of untreated and treated with different concentration of RSL3 and erastin with or without Lip-1. Scale bar = 100 μm. (**C**,**D**) Dose-dependent toxicity of oxidative cell death-inducing agents (RSL3 and erastin). Both cell lines were treated with increasing concentrations of GPX4 inhibitor RSL3 and system xCT inhibitor erastin with or without Lip-1. Data shown represent the mean ± s.d. * *p* < 0.05, ** *p* < 0.01, **** *p* < 0.0001. (two-way ANOVA), of *n* = 3 wells of a 96-well plate, from three independent experiments. Cell viability was measured after 24 h of treatment.

**Figure 4 biology-13-00215-f004:**
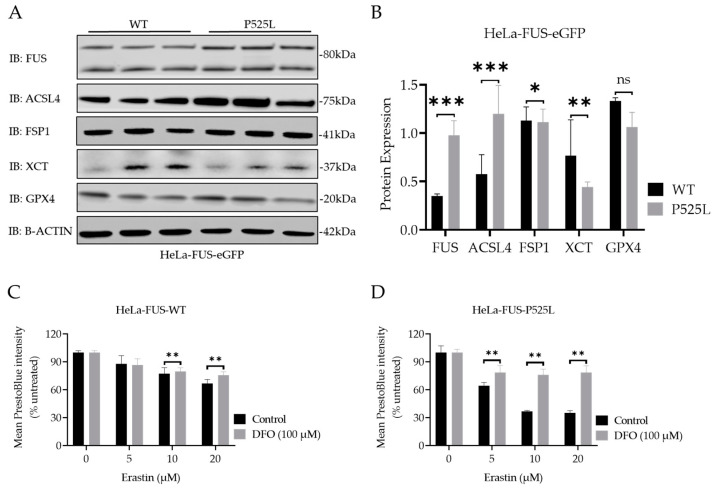
ALS causing FUS mutation leads to misregulation of the protein expression of established ferroptotic factors. (**A**) Immunoblot analysis of both HeLa-FUS-WT and HeLa-FUS-P525L cell lines of FUS, GPX4, XCT, FSP1, and ACSL4 using specific antibodies. See also Appendix A. (**B**) Quantification of the expression of the key ferroptotic proteins normalized to the reference gene B-Actin. Data shown from three independent experiments represent the mean ± s.d. ns = non-significant * *p* < 0.05, ** *p* < 0.01, *** *p* < 0.001; (two-way ANOVA), of *n* = 3. (**C**,**D**) Both cell lines were treated with increasing concentrations of erastin with or without the ferroptosis inhibitor DFO (100 µM). Cell viability was analyzed using Presto Blue method 24 h after the treatment. Data shown represent the mean ± s.d. ** *p* < 0.01; (two-way ANOVA), of *n* = 3 wells of a 96-well plate, from three independent experiments.

**Figure 5 biology-13-00215-f005:**
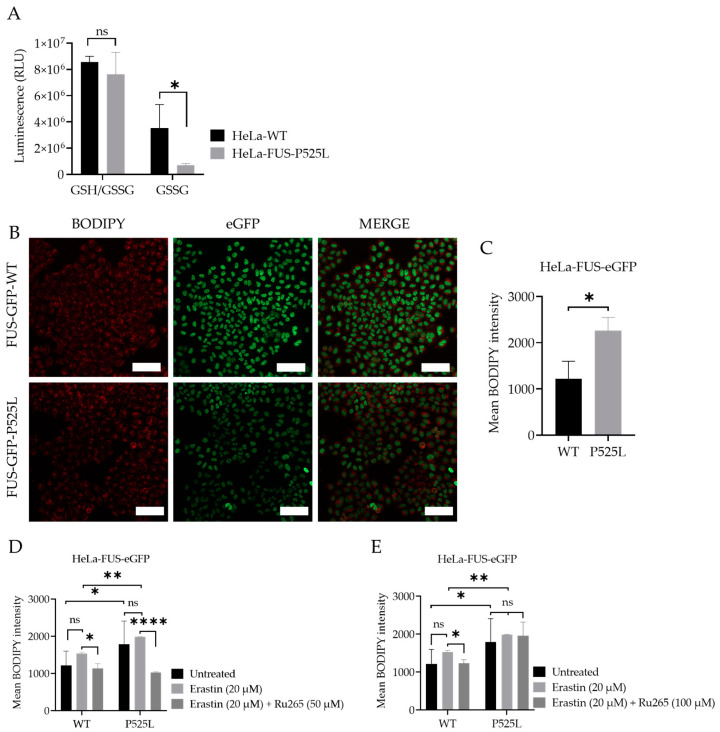
Cellular redox defense system is impaired by FUS mutations, which can be mitigated by mitochondrial calcium uniporter inhibition. (**A**) Luciferase-based assay used to analyze total glutathione (GSH) and oxidized glutathione (GSSG) levels in both cell lines. Data shown represent the mean ± s.d. ns > 0.9999, * *p* < 0.05; (two-way ANOVA), normalized with vehicle (untreated control) of *n* = 3 wells of a 96-well plate, from three independent experiments. (**B**) Live-cell fluorescence imaging of lipid peroxides in HeLa-FUS-WT and HeLa-FUS-P525L. Scale bars = 100 µm. (**C**) Lipid peroxides florescence intensity relative to baseline without any treatment in HeLa-FUS-WT compared to HeLa-FUS-P525L. (**D**,**E**) Lipid peroxides fluorescence intensity after 20 µM erastin treatment for 24 h with and without Ru265 (50 µM) and Ru265 (100 µM) in both HeLa-FUS-WT and HeLa-FUS-P525L. Control untreated (no erastin and no Ru265). Data are from a minimum of 3 stage positions. For statistical analysis, each stage position counted as one data entry. Data are mean ± s.d., * *p* < 0.05, ** *p* < 0.01, **** *p* < 0.0001; (two-way ANOVA), of *n* = 3 wells of a 96-well plate, from three independent experiments.

**Figure 6 biology-13-00215-f006:**
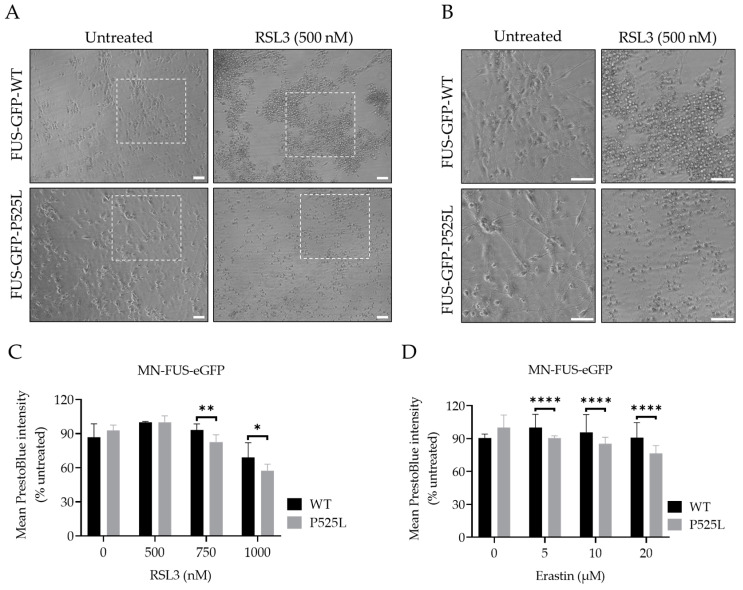
Human-induced pluripotent stem cell-derived neural progenitor cell (NPC)-generated motor neurons (MN) show higher susceptibility to ferroptosis. (**A**) Brightfield images of both FUS- WT and FUS-P525L smNPCs-generated MN with and without treatment of RSL3. Scale bar = 100 pixels. (**B**) is the magnification area of A. (**C**,**D**) Dose-dependent toxicity of oxidative cell death. Both cell lines were treated with increasing concentrations of RSL3 and erastin. Data shown represent the mean ± s.d. * *p* < 0.05, ** *p* < 0.01, **** *p* < 0.0001; (two-way ANOVA), of *n* = 3 wells of a 96-well plate, from three independent experiments. Cell viability was measured after 48 h.

## Data Availability

All data are presented in this manuscript.

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
