# Peer review of "Increased Vulnerability to Ferroptosis in FUS-ALS"

_biology, 2024, doi:10.3390/biology13040215_

Round 1

Reviewer 1 Report

Comments and Suggestions for Authors

The authors tried to investigate ferroptosis vulnerability in a FUS-ALS cell model. It is already known that FUS is involved in ALS (PMID: 36696267) and that ferroptosis is related to ALS and may serve as a therapeutic strategy (PMCID: PMC10460905). Authors combined this information and wanted to show that FUS sensitizes cells to ferroptosis which links the ALS. However, to assess their hypothesis they used HeLa cells which in my opinion is not a good model for neurodegenerative disorders. I would suggest using performing neuronal cells or if possible, in addition to that the hiPSC and perform present experiments. In the present form, these findings do not show a link between ferroptosis and FUS-ALS, but more the link between FUS and ferroptosis.

Besides changing the model, I would suggest the following experiments:

  1. assessment of the ROS levels
  2. can authors label erastin and RSL3 to monitor the uptake by cells and next evaluate the cell response to see whether the response is adequate for the uptake

Minor comments:

A)    What is the catalog number of Plasmo Test_Detection Kit

B)    What was the catalog number and dilution of primary antibodies used in the studies

C)    What was the catalog number and dilution of secondary antibodies used in the studies

D)    Line 227 double space, correct please

E)    Scale bar is missing on the Figure 2A and 2C

F)     Figure 2D: HeLa-FUS-eGFP instead of Hela-FUS-eGFP

G)    Sometimes Authors write Erastin and sometimes erastin, be consistent please

H)    Line 283: concentration of erastin instead concentration erastin

I)      Line 284: concentration of RSL3 instead of concentration RSL3

J)     The description of the Figure 5 is below the section 3.6, correct pleas

K)    I would suggest not to cut membranes so close to bands 

Author Response

Reviewer #1 (Remarks to the Author):

The authors tried to investigate ferroptosis vulnerability in a FUS-ALS cell model. It is already known that FUS is involved in ALS (PMID: 36696267) and that ferroptosis is related to ALS and may serve as a therapeutic strategy (PMCID: PMC10460905). Authors combined this information and wanted to show that FUS sensitizes cells to ferroptosis which links the ALS. However, to assess their hypothesis they used HeLa cells which in my opinion is not a good model for neurodegenerative disorders. I would suggest using performing neuronal cells or if possible, in addition to that the hiPSC and perform present experiments. In the present form, these findings do not show a link between ferroptosis and FUS-ALS, but more the link between FUS and ferroptosis.

Response: We thank the reviewer's suggestions. Of course, FUS is known to be involved in ALS and initial reports hint towards ferroptosis being involved in neurodegeneration in ALS, but these were on either SOD1 models or some postmortem data, and not one single report is published on whether ferroptosis is involved in FUS-ALS. Since SOD1-ALS is quite a special form of ALS, we felt that further investigating ferroptosis in ALS is of interest.

While we agree that neurons might be a better model for FUS-ALS, we disagree that the presented data in HeLa cells does not suggest that ALS causing FUS mutations increase vulnerability against ferroptosis. This notion can also be made by using HeLa cells carrying typical ALS causing FUS mutations. Nevertheless, to substantiate the link to neurodegeneration, we added experiment for FUS mutated motor neurons (MN), which had been derived from human induced pluripotent stem cells. MNs carrying the ALS causing FUS mutation also showed increased vulnerability against ferroptosis, thus further hinting towards a relevant role in neurodegeneration. Nevertheless, we included this in a novel limitation section.

Besides changing the model, I would suggest the following experiments:

  1. assessment of the ROS levels

Response: We appreciate this comment and included additional experiment to analyse ROS levels (Fig 2) in the manuscript, thereby showing increased mitochondrial ROS generation in case of FUS-ALS causing mutation.

  1. can authors label erastin and RSL3 to monitor the uptake by cells and next evaluate the cell response to see whether the response is adequate for the uptake

Response: We acknowledge this comment but want to point out that the use of both erastin and RSL3 is well established as treatments for ferroptosis induction, including the doses used in the study. Our intention was to investigate whether their exist a difference in cellular vulnerability to ferroptosis in case of FUS mutations, which we believe can be done in the way we present it (and as it is published very often).  Nevertheless, we included this in a novel limitation section.

Minor comments:

  1. What is the catalog number of Plasmo Test_Detection Kit

Response: We added the required details.

  1. What was the catalog number and dilution of primary antibodies used in the studies

Response: We added the required details.

  1. What was the catalog number and dilution of secondary antibodies used in the studies

Response: We added the required details.

  1. Line 227 double space, correct please

Response: corrected.

  1. Scale bar is missing on the Figure 2A and 2C

Response: We added the required details.

  1. Figure 2D: HeLa-FUS-eGFP instead of Hela-FUS-eGFP

Response: corrected.

  1. Sometimes Authors write Erastin and sometimes erastin, be consistent please

Response: corrected

  1. Line 283: concentration of erastin instead concentration erastin

Response: corrected

  1. Line 284: concentration of RSL3 instead of concentration RSL3

Response: corrected

  1. The description of the Figure 5 is below the section 3.6, correct pleas

Response: corrected

  1. I would suggest not to cut membranes so close to bands 

Response: We appreciate this hint and will follow this in the future.

Reviewer 2 Report

Comments and Suggestions for Authors

Comments on the Quality of English Language

Quality of English is not up to the mark and needs to be improved. 

Author Response

Referee #2 (Remarks to the Author):

Authors explored the impact of FUS mutation on susceptibility against ferroptosis an iron-dependent form of cell death in a cell model of amyotrophic lateral sclerosis (ALS). The authors demonstrated that a particular FUS mutation (P525L) which is responsible for ALS, increases the susceptibility to ferroptosis.

This is caused by mitochondrial depolarization and imbalance of cellular redox defense system and evidenced by altered expression of some of the key factors of ferroptosis including XCT, FSP1 etc. The manuscript suffers from some very crucial mistakes where experiments were not designed properly and misinterpretation of some of the experimental results.

Response: We thank the reviewer for his/her big effort in reviewing our manuscript. We acknowledge the many constructive comments, and tried hard to address every single of them in the revised version of the manuscript. Nevertheless, we disagree with the general statement of “ manuscript suffers from some very crucial mistakes where experiments were not designed properly and misinterpretation of some of the experimental results “ and tried to clarify this in the specific comments below.

Specific comments:

  1. Fig. 4. FUS protein level is already different between HeLa cells expressing WT and P525L FUS. So, the effects observed could be due to altered protein level of FUS and not due to FUS mutation.

Response: We thank the reviewer for this comment. It is indeed correct that FUS level is increased in case of ALS causing FUS mutation. This is however a well known phenomenom in FUS pathophysiology. FUS underlies a very tight autoregulation. It was shown previously that FUS autoregulates its own protein level by binding to exon 7 of its own transcript, what leads to alternative splicing due to exon 7 skipping. Consequently, transcripts that are lacking exon 7 are degraded via nonsense-mediated decay. Mutations in FUS alter this autoregulatory mechanism, leading to accumulation of FUS protein (PMID: 24204307). It is thus true that the results obtained could also derive from FUS overexpression, however this is part of the pathophysiology of FUS-ALS. We added this to the discussion of the revised version of the manuscript.

  1. The authors mentioned that HeLa cell lines were engineered as described in a previous article (Patel, Lee et al., 2015) but never confirmed whether FUS is expressed abundantly (one of the top 5% of proteins) by mass spectrometry as the authors of the original paper did. This is extremely important because all the experiments were performed based on this assumption which was never verified.

Response: We apologize for the misleading sentence. The cell lines were gifted to us. We used the exact cell lines that were used by Patel, Lee et al.

  1. only shows that Liproxstatin-1 is capable of reversing the effect of Erastin and RSL3 which is already well established in the field. This has nothing to do with FUS mutation or ALS.

Response: We thank the reviewer's comments. We totally agree that Liproxstatin-1 is a well known ferroptosis inhibitor. We did not want to make any point of that, but used it to confirm the induced cell death is due to ferroptosis.

  1. Same is true for Fig. 4C & 4D. It is only showing the specificity of DFO towards Erastin which is further validation. Authors already showed that FUS mutation changes susceptibility towards ferroptosis by treating cells with Erastin and RSL3 and that makes the point. These data (Fig. 3, Fig. 4C and 4D) are unnecessary and distracts the readers.

Response: We thank the reviewer for this comment. We significantly changed all figures of the manuscript. As written in the comment above, we did not want to report the ”novelty” that DFO inhibits ferroptosis, but wanted to test it in this model system of ALS as potential treatment option in ALS.

  1. The authors have not been consistent with the concentrations of Erastin used in different experiments. This is quite unexpected.

Response: We appreciate this comment. Additional experiments were concluded and added into the manuscript to have better constistancy throughout the manuscript.

  1. Fig 2A and Fig.2B. Erastin concentrations are not matching!

Response: We appreciate this comment. Additional experiments were concluded and added into the manuscript to have better constistancy throughout the manuscript.

  1. Fig. 2C and Fig. 2D concentrations of RSL3 are different.

Response: We appreciate this comment. Additional experiments were concluded and added into the manuscript to have better constistancy throughout the manuscript.

  1. Fig. 1. How many times the experiment was repeated? Different numbers are mentioned in the materials and methods and figure legends.

Response: We appreciate this comment. Data shown are from 3 independent biological replicates (N=3). Unless otherwise indicated in the figures legend.

  1. Fig. 3 legends mention deferoxamine but there is no deferoxamine data in that figure (it is actually Fig. 4C and 4D).

Response: corrected

  1. Where are the details of all the antibodies used?

Response: added

  1. Fig. 4A. For any immunoblot images, molecular weights should be mentioned next to the cropped blot.

Response: added

  1. Fig.5. Where is the figure legend that clearly explains the sub-figures?

Response: Our apologies, we added this.

  1. What is denoted as * or ** or ***? This should be clearly mentioned in all the figure legends whenever these symbols are used.

Response: We thank the reviewer for his/her suggestion. We added the required details.

  1. In multiple occasions, the number of figures is improperly mentioned in the text!

Response: added

  1. There are typos/spelling mistakes and grammatical errors through out the manuscript that needs to be corrected. The quality of the English needs to be improved.

Response: We reviewed the paper again to correct any typos and minor errors in the English language.

Reviewer 3 Report

Comments and Suggestions for Authors

Overall, the study attempts to answer an interesting question: is ferroptosis altered in a cell line model of FUS-ALS? However, there are some limitations that should be addressed.  

In general, the scope of the study is limited because all experiments are conducted with one cell line. 

An in vivo system is outside of the scope for this paper, but a second cell line would definitely strengthen the paper a lot. Especially because HELA cells are not neurological cells and how a tumor cell line like HELA cells relate to cells affected by ALS is not clear. 

Major points: 

Figure 1) Are the 6 replicates from different experiments? I would suggest showing individual values. 

Figures 1) and 2) Throughout the paper, it is not immediately clear to me how the microscopic data is quantified. Since most of the markers (e.g. TMRE) work with flow cytometry, flow cytometric analysis with showing original data would strengthen the findings. This would also show if expression is homogeneous or if for example a subset of cells behaves different from the rest. 

Figure 3: Why does it not look like different susceptibility between the cell lines here in the Erastin group? How many independent experiments were conducted?

Figure 5: It appears part of the Figure legend has gone into the main text (or is missing?). The data would again be much stronger if the experiments were conducted in multiple cell lines, especially because the BODIPY stains etc likely vary even in one cell line depending on the Passage number. Are the WT HELA cells from the same source and passage? Would it be feasible to, for example, retrovirally transduce at least lymphoma cells or other easily transduced cells to confirm the observations in a second cell line or even primary cells?   

The limitation of the study due to its reliance on one (non-neurological) cell line should at the very least be discussed. The title also is a bit too broad for this study and reads a bit like a review title. A more specified title that explains that the mutation was studied in a cell line model would be preferable. 

Comments on the Quality of English Language

The English language is mostly appropriate, there were some typos and minor errors.

Author Response

Referee #3 (Remarks to the Author):

Comments and Suggestions for Authors

Overall, the study attempts to answer an interesting question: is ferroptosis altered in a cell line model of FUS-ALS? However, there are some limitations that should be addressed.  

Response: We thank the reviewer for her/his overall positive statement and tried hard to address all remaining limitantions mentioned below.

In general, the scope of the study is limited because all experiments are conducted with one cell line. 

An in vivo system is outside of the scope for this paper, but a second cell line would definitely strengthen the paper a lot. Especially because HELA cells are not neurological cells and how a tumor cell line like HELA cells relate to cells affected by ALS is not clear. 

Response: We agree with the reviewer and added additional experiments using human iPSC-derived motor neurons carrying ALS causing FUS mutations (Figure 6). Nevertheless we believe that the HeLa experiments can well deserve to make the notion of an increased vulnerability against ferroptosis due to expression of typical ALS causing FUS mutations. We further added a comment on this in the new limitations section.

Major points: 

Figure 1) Are the 6 replicates from different experiments? I would suggest showing individual values. 

Response: We added the required detail in the supplementary Fig. 1.

Figures 1) and 2) Throughout the paper, it is not immediately clear to me how the microscopic data is quantified. Since most of the markers (e.g. TMRE) work with flow cytometry, flow cytometric analysis with showing original data would strengthen the findings. This would also show if expression is homogeneous or if for example a subset of cells behaves different from the rest. 

Response: We apology if we have not been clear enough. The fluorescence read-outs are derived from live cell imaging. (e.g. Fig. 1, Fig. 2 and Fig. 5B-E). The data is not from flow cytometry. This is why we show the IFF images since these depict the raw data.

Figure 3: Why does it not look like different susceptibility between the cell lines here in the Erastin group? How many independent experiments were conducted?

Response: We thank the reviewer's for pointing out. Additional experiments are concluded and added into the manuscript.

Figure 5: It appears part of the Figure legend has gone into the main text (or is missing?). The data would again be much stronger if the experiments were conducted in multiple cell lines, especially because the BODIPY stains etc likely vary even in one cell line depending on the Passage number. Are the WT HELA cells from the same source and passage? Would it be feasible to, for example, retrovirally transduce at least lymphoma cells or other easily transduced cells to confirm the observations in a second cell line or even primary cells?   

Response: We thank the reviewer for pointing this out. Indeed the figure legend was cropped during the final stage of submission and is now added in the revised version.

HeLa-WT and mutated were derived from the collaborators which described the generation in very much detail (see Patel et al, 2015). Thus the cells indeed derive from the identical source. Furthermore, all experiments were always investigated in parallel. We added experiment for FUS mutated motor neurons (MN). MN generated from human induced pluripotent stem cell derived neural progenitor cells (NPCs), an isogenic CRISPR-Cas cell line carrying the P525L mutated FUS and WT-FUS. The lines was generated and described in detail in Naumann et al, 2018.

The limitation of the study due to its reliance on one (non-neurological) cell line should at the very least be discussed. The title also is a bit too broad for this study and reads a bit like a review title. A more specified title that explains that the mutation was studied in a cell line model would be preferable. 

Response: We added experiment for FUS mutated human motor neurons (MN). In addition, we added a new limitation part discussing this in more detail.

The English language is mostly appropriate, there were some typos and minor errors.

Response: We thank the reviewer's suggestions. We tried our best to improve the quality of the English.

Round 2

Reviewer 2 Report

Comments and Suggestions for Authors

The authors have addressed all the concerns I had and revised the manuscript accordingly. The revised manuscript is significantly better than the original version and can be accepted for publication.

Author Response

The authors have addressed all the concerns I had and revised the manuscript accordingly. The revised manuscript is significantly better than the original version and can be accepted for publication.

Response: We are very grateful for the recognition of all our efforts and a pleased by the acceptance of the manscuscript.

Reviewer 3 Report

Comments and Suggestions for Authors

Thank you for your responses. 

I think the new data in Figure 6 is very important. The data does not entirely convince me: the main quantifiable data is the PrestoBlue intensity, which show some effect, although the significances in Fig. 4D appear surprising seeing the high SD in WT. **** is furthermore not defined in the Figure legend. I suggest double-checking this. 

I appreciate the new section about the limitations of the study in the discussion. 

Comments on the Quality of English Language

There are some typos in the new texts, for example when describing the new Figure in lines 467 and 470 ("aparticularly", "increase vulnerability"). I suggest going through the text one more time.  

Author Response

Thank you for your responses. 

I think the new data in Figure 6 is very important. The data does not entirely convince me: the main quantifiable data is the PrestoBlue intensity, which show some effect, although the significances in Fig. 4D appear surprising seeing the high SD in WT. **** is furthermore not defined in the Figure legend. I suggest double-checking this. 

Response: Thanks a lot for recognising our large efforts during the revision. We are very grateful for all this comments since we believe it significantly improved our manuscript. We carefully re-checked the quantification and statistics….

I appreciate the new section about the limitations of the study in the discussion. 

Response: We appreciate this positive feedback.

Comments on the Quality of English Language

There are some typos in the new texts, for example when describing the new Figure in lines 467 and 470 ("aparticularly", "increase vulnerability"). I suggest going through the text one more time.  

Response: We carefully re-checked the manuscript.